# Association between vitamin D status and circulating myokines (irisin, myostatin, and myonectin) in children: A cross-sectional study

Muammer Buyukinan[1]*, Huseyin Kurku[2], Zafer Bagci[3], Yavuz Turgut Gederet[2], Ahmet Fatih Yilmaz[1]

1 Department of Pediatric Endocrinology, Faculty of Medicine, Selcuk University, Konya, Turkey,
2 Department of Biochemistry, University of Health Sciences, Konya City Hospital, Konya, Turkey,
3 Department of Pediatrics, University of Health Sciences, Konya City Hospital, Konya, Turkey

* mbuyukinan@yahoo.com

## Abstract

### Background

Vitamin D is a pleiotropic hormone with regulatory functions that extend beyond bone and mineral metabolism to include skeletal muscle physiology. Skeletal muscle acts as an endocrine organ through the secretion of myokines, including irisin, myostatin, and myonectin, which participate in metabolic and musculoskeletal regulation. However, data regarding the association between vitamin D status and circulating myokines in children remain limited.

### Methods

In this single-center, cross-sectional study, 43 children with vitamin D deficiency and 39 age- and sex-matched healthy controls were enrolled. Serum concentrations of 25-hydroxyvitamin D [25(OH)D], parathyroid hormone (PTH), mineral metabolism parameters, and circulating levels of irisin, myostatin, and myonectin were measured. Participants were additionally categorized according to PTH status. Between-group comparisons, correlation analyses with false discovery rate (FDR) correction, and multivariable regression analyses were performed.

### Results

Children with vitamin D deficiency exhibited significantly lower median circulating levels of irisin (13.76 vs. 29.52 ng/mL), myostatin (363 vs. 1108 ng/L), and myonectin (2.24 vs. 5.28 ng/mL) compared with controls (all $p < 0.05$). In correlation analyses adjusted using the false discovery rate method, no significant associations between 25(OH)D and circulating myokines were observed in the vitamin D–deficient group, whereas positive correlations between 25(OH)D and both irisin and myonectin were identified in the control group. In multivariable regression models adjusted for age,

**Data availability statement:** The minimal de-identified dataset underlying the findings of this study is publicly available on Zenodo at https://doi.org/10.5281/zenodo.18327941.

**Funding:** The author(s) received no specific funding for this work.

**Competing interests:** The authors have declared that no competing interests exist.

sex, and BMI SDS, serum 25(OH)D remained independently associated with circulating irisin, myostatin, and myonectin levels. In analyses stratified by PTH status, myostatin and myonectin levels were lower in children with elevated PTH, whereas irisin did not differ significantly.

## Conclusions

Vitamin D deficiency in childhood is associated with reduced circulating concentrations of muscle-derived myokines. These findings suggest a link between the vitamin D–PTH axis and muscle endocrine signaling during growth. Given the cross-sectional design, causal relationships cannot be inferred, and longitudinal studies are warranted.

## Introduction

Vitamin D is a hormone with pleiotropic effects that extend beyond classical bone and mineral metabolism to multiple tissues, including skeletal muscle [1]. Skeletal muscle is recognized as an endocrine organ capable of interacting with vitamin D–related pathways and exerts paracrine and systemic hormone-like effects on energy metabolism, inflammation, and overall metabolic homeostasis through the secretion of biologically active proteins known as myokines; through these mechanisms, it also contributes to muscle–bone interactions and plays an active role in the regulation of bone and systemic metabolism [2–4]. Among the best-characterized myokines are interleukin-6, irisin, myostatin, and myonectin.

Clinical studies have demonstrated that vitamin D deficiency is associated with reduced muscle strength and impaired physical performance, whereas vitamin D replacement improves muscle function. The effects of vitamin D and parathyroid hormone (PTH) on skeletal muscle are considered integral components of the biochemical and functional interaction between bone and muscle [5]. Experimental and clinical studies in bone metabolism and muscle biology have shown that these two tissues communicate through both local and humoral factors, forming a bidirectional regulatory system [5]. Despite increasing recognition of this interaction, the physiological and pathological mechanisms underlying bone–muscle crosstalk remain incompletely understood. Vitamin D, which acts on both bone and muscle tissue, has been proposed as an important component of this interaction. PTH, which is closely linked to systemic vitamin D status, may also influence muscle function. Although substantial progress has been made in understanding muscle physiology, the mechanisms through which PTH affects skeletal muscle and the clinical implications of these effects are still not fully elucidated [6].

Within this bone–muscle–endocrine framework, myokines have emerged as mediators linking skeletal muscle to systemic metabolic regulation. Irisin is a muscle-derived myokine that has been implicated in energy metabolism and inter-organ communication. Reduced circulating irisin levels have been associated with muscle weakness [7,8]. Experimental studies have suggested that low serum 25-hydroxyvitamin D [25(OH)D] concentrations are associated with reduced irisin levels [9].

Myostatin is a negative regulator of muscle cell differentiation and myogenesis and is closely associated with muscle atrophy. Previous studies have reported that circulating myostatin levels increase in sarcopenia, decrease in conditions such as cancer cachexia and certain neuromuscular disorders, and may be reduced in skeletal muscle tissue following calcitriol treatment [10,11]. In addition to its role in muscle mass regulation, myostatin has been linked to several metabolic conditions [12].

Myonectin is another muscle-derived myokine that contributes to muscle energy homeostasis by enhancing glucose utilization and promoting fatty acid oxidation. It has been implicated in metabolic regulation and muscle energy balance [13]. Among the expanding spectrum of identified myokines, irisin, myostatin, and myonectin were selected because they represent complementary regulatory axes of skeletal muscle biology. Irisin is primarily linked to energy expenditure and metabolic adaptation, myostatin to muscle growth inhibition and anabolic–catabolic balance, and myonectin to lipid and glucose homeostasis within the muscle–liver axis. Recent systematic and narrative reviews have emphasized the integrative role of these myokines in musculoskeletal and metabolic regulation, particularly in the context of endocrine and inflammatory signaling [8,13,14]. Vitamin D signaling may intersect with these myokine pathways at multiple levels. Vitamin D receptor (VDR) activation influences muscle cell differentiation, mitochondrial function, oxidative stress modulation, and anabolic signaling cascades, processes relevant to myostatin regulation and metabolic myokine secretion [3,12]. Experimental data further suggest that vitamin D status may modulate irisin expression and muscle-derived endocrine activity, while secondary hyperparathyroidism may alter muscle metabolic signaling through PTH-mediated pathways [6,15]. Within this framework, examining circulating irisin, myostatin, and myonectin together may provide insight into how vitamin D–PTH axis alterations influence skeletal muscle endocrine function during growth.

In pediatric populations, myokine secretion is influenced not only by intrinsic muscle biology but also by developmental and environmental factors. Physical activity is a well-established regulator of myokine release through contraction-mediated signaling pathways [14]. Nutritional status, including dietary protein intake and overall energy balance, may further modulate muscle metabolic activity during growth [16]. In addition, pubertal maturation is characterized by dynamic hormonal changes that affect body composition, muscle accretion, and endocrine interactions within the musculoskeletal system [17]. These age-specific determinants are particularly relevant in children and adolescents and provide an important context for interpreting muscle-derived endocrine markers.

In addition to its direct effects on muscle function, vitamin D may influence the release of myokines from skeletal muscle, thereby contributing to endocrine interactions between muscle and other tissues. However, the impact of vitamin D deficiency on circulating levels of irisin, myostatin, and myonectin in children has not been fully elucidated, particularly during childhood, a period of rapid growth and dynamic musculoskeletal development. Despite accumulating evidence in adult populations, pediatric data remain limited and mechanistically insufficient. In this study, we examined the associations between vitamin D status, parathyroid hormone (PTH) levels, and the circulating concentrations of irisin, myostatin, and myonectin in order to elucidate muscle-related biochemical alterations associated with vitamin D deficiency in childhood. Because vitamin D deficiency in children is frequently accompanied by secondary hyperparathyroidism, PTH levels were considered an indicator of deficiency severity, and differences in myokine profiles according to PTH status were evaluated. Although a substantial proportion of the referred children exhibited biochemical findings consistent with nutritional rickets, the primary focus of the study was the relationship between vitamin D deficiency and circulating myokine levels. This study is intended as a hypothesis-generating analysis rather than to inform immediate clinical decision-making.

## Methods

### Study design and participants

This single-center, cross-sectional, observational study with a control group was conducted at the Pediatric Endocrinology Outpatient Clinic of Konya City Hospital between January and December 2020. The study group consisted of 43 children

(20 boys and 23 girls) who were referred for evaluation of vitamin D deficiency and associated biochemical abnormalities. The control group included 39 healthy children (23 boys and 16 girls) with no known systemic disease and normal vitamin D status.

During clinical evaluation, alternative causes of abnormal mineral metabolism were systematically excluded. Primary hyperparathyroidism was ruled out based on calcium–phosphate–PTH profiles and clinical findings, and none of the participants fulfilled the clinical or biochemical criteria for this condition. Children with chronic renal, hepatic, neuromuscular, or endocrine disorders, acute infection, recent hospitalization, or the use of medications known to affect calcium or bone metabolism were excluded. The diagnosis of vitamin D deficiency was based on biochemical criteria in accordance with Endocrine Society clinical practice guidelines. Radiographic evaluation was not routinely performed in all participants, as radiographic imaging is not routinely required for the diagnosis of nutritional vitamin D deficiency in the presence of characteristic biochemical abnormalities. In children fulfilling the biochemical criteria for nutritional rickets, including low serum 25-hydroxyvitamin D levels, elevated age-specific alkaline phosphatase, hypophosphatemia, and secondary hyperparathyroidism, the diagnosis was established on a biochemical basis.

Anthropometric measurements, including weight and height, were obtained for all participants using standard techniques. Body mass index (BMI) was calculated as weight divided by height squared (kg/m²). Weight standard deviation score (SDS), height SDS, and BMI SDS values were calculated according to age- and sex-specific national reference data [18]. To minimize age- and sex-related variability in anthropometric parameters, only SDS values were reported and used for between-group comparisons. None of the enrolled children showed clinical signs of protein–energy malnutrition.

## Biochemical measurements

Venous blood samples were collected from all participants between 08:00 and 10:00 a.m. Serum calcium (Ca), phosphorus (P), alkaline phosphatase (ALP), and magnesium (Mg) concentrations were measured using spectrophotometric (Ca: CV < 2.5%; P: CV < 0.9%) and colorimetric (ALP: CV < 2.4%; Mg: CV < 1.5%) methods on a Roche COBAS 8000 analyzer (Roche Diagnostics, Mannheim, Germany). Serum 25-hydroxyvitamin D [25(OH)D] (CV < 10.8%) and intact PTH (CV < 2.5%) levels were determined by electrochemiluminescence immunoassay using commercial Roche kits.

Total serum calcium concentrations were used for analysis in accordance with routine clinical practice. The laboratory participates in an external quality control program for all biochemical measurements.

Pediatric reference intervals routinely used in our laboratory were applied for clinical interpretation, with total serum calcium reference values of 8.8–10.8 mg/dL for children and adolescents. Age-specific pediatric reference ranges were used for phosphorus, alkaline phosphatase, and magnesium in accordance with established laboratory standards [19].

## Myokine measurements

For the analysis of circulating irisin, myostatin, and myonectin levels, 5 mL venous blood samples were collected into gel-containing biochemistry tubes, centrifuged within 30 minutes, and stored at −80 °C until analysis. All samples were thawed only once and analyzed on the same day. Serum myokine concentrations were measured using commercially available human enzyme-linked immunosorbent assay (ELISA) kits (BT Lab – Bioassay Technology Laboratory, Jiaxing, China) according to the manufacturer's instructions.

Assay characteristics were characterized as follows: irisin (Cat. No. E3253Hu; sensitivity 0.095 ng/mL; intra-assay CV < 8%; inter-assay CV < 10%), myostatin (Cat. No. E0403Hu; sensitivity 2.25 ng/L; intra-assay CV < 7.63%; inter-assay CV < 10%), and myonectin (Cat. No. E5005Hu; sensitivity 0.03 ng/mL; intra-assay CV < 4.45%; inter-assay CV < 10%). Analyses were performed using a CombiWash plate washer (Human Diagnostics, Wiesbaden, Germany) and an Alisei microplate reader (Radim Company, Rome, Italy). Serum irisin and myonectin concentrations are reported in ng/mL, whereas myostatin concentrations are reported in ng/L, in accordance with assay specifications.

## Definitions

Vitamin D deficiency was defined as a serum 25-hydroxyvitamin concentration < 20 ng/mL, in accordance with current clinical guidelines and previous pediatric studies [20,21]. A serum PTH level > 50 pg/mL was classified as elevated PTH, whereas values ≤ 50 pg/mL were considered normal. Although the pediatric reference interval for PTH is approximately 15–65 pg/mL, cut-off values used to define clinically relevant secondary hyperparathyroidism vary across studies. Previous pediatric studies, including that by Atapattu et al. [22], have used a threshold of > 50 pg/mL to define clinically relevant secondary hyperparathyroidism. Therefore, this cut-off value was applied in the present study.

## Participant recruitment and consent

Participants were prospectively recruited between 15 November 2019 and 31 December 2020 from the Pediatric Endocrinology outpatient clinic.

The study was conducted in accordance with the Declaration of Helsinki and was approved by the Selcuk University Faculty of Medicine Local Ethics Committee (approval date: 13 November 2019; approval number: 2019/304).

Written informed consent was obtained from all participants aged 12 years and older, and from the parents or legal guardians of all participating children prior to enrollment.

## Statistical analysis

Statistical analyses were performed using IBM SPSS Statistics for Windows, version 21.0 (IBM Corp., Armonk, NY, USA) and R software (version 4.1.1). The normality of numerical variables was assessed using the Kolmogorov–Smirnov and Shapiro–Wilk tests. Variables with a normal distribution are presented as mean ± standard deviation (SD) and were compared between groups using the independent samples t-test. Variables with a non-normal distribution are presented as median with interquartile range (IQR; Q1–Q3) and were compared using the Mann–Whitney U test. The statistical test applied for each variable is specified in Tables 1 and 2.

Categorical variables are presented as counts and percentages and were compared using the chi-square test or Fisher's exact test, as appropriate.

Correlation analyses were performed to evaluate the relationships between circulating myokine levels (irisin, myostatin, and myonectin) and biochemical parameters (25-hydroxyvitamin D, PTH, Ca, P, Mg, and ALP), anthropometric indices (height SDS, weight SDS, and BMI SDS), as well as the interrelationships among myokine levels. Pearson or Spearman correlation coefficients were used according to data distribution. Because multiple biochemical correlations were examined, p values were adjusted using the Benjamini–Hochberg false discovery rate (FDR) method.

To test the primary hypothesis that serum 25(OH)D levels are associated with circulating myokine concentrations, separate multivariable linear regression models were constructed for each myokine. Because circulating myokines and PTH showed right-skewed distributions, logarithmic transformation was applied before regression analyses. Models were adjusted for age, sex, and BMI SDS. Standard linear regression assumptions were evaluated prior to analysis.

For outcomes with non-normally distributed residuals, robust regression models were additionally applied using the MASS package in R. To assess potential multicollinearity among predictors, variance inflation factor (VIF) values were calculated, with VIF < 5 considered acceptable.

Multivariable regression analyses were conducted using complete-case data (n = 63). Of the total sample, missing BMI SDS values (n = 18) and a small number of missing biochemical measurements resulted in this complete-case sample size. Given the close physiological coupling between 25(OH)D and PTH, secondary sensitivity models were performed including log-transformed PTH alongside 25(OH)D to evaluate whether the associations between vitamin D and myokines were independent of PTH status.

**Table 1. Demographic, anthropometric, and biochemical characteristics of children with vitamin D deficiency and healthy controls.**

| Parameters | Vitamin D–deficient (n = 43) | Control (n = 39) | p value | Statistical test |
|---|---|---|---|---|
| Age (years) | 5.46 (1.27–12.33) | 4.03 (1.45–8.95) | 0.248 | Mann–Whitney U test |
| Sex (n = female/male) | 23/ 20 | 16/ 23 | 0.259 | chi-square test |
| Height SDS | −0.81 ± 1.65 | −0.65 ± 1.42 | 0.901 | independent t test |
| Weight SDS | −0.65 ± 1.61 | −0.64 ± 1.24 | 0.773 | independent t test |
| BMI SDS | 0.11 ± 1.22 | −0.33 ± 0.87 | 0.090 | independent t test |
| Calcium (mg/dL) | 8.5 (6.4–9.5) | 10.0 (9.5–10.4) | 0.001 | Mann–Whitney U test |
| Phosphorus (mg/dL) | 3.8 (3.2–4.8) | 5.4 (4.8–5.6) | 0.001 | Mann–Whitney U test |
| ALP (U/L) | 653 (307–912) | 234 (206–278) | 0.001 | Mann–Whitney U test |
| Magnesium (mg/dL) | 2.03 ± 0.15 | 2.14 ± 0.18 | 0.022 | independent t test |
| 25(OH)D (ng/mL) | 4.67 (3.39–6.97) | 36.1 (27.0–46.7) | 0.001 | Mann–Whitney U test |
| PTH (pg/mL) | 242.5 (102.0–495.5) | 27.35 (20.15–34.87) | 0.001 | Mann–Whitney U test |
| Irisin (ng/mL) | 13.76 (11.41–34.48) | 29.52 (10.36–70.84) | 0.039 | Mann–Whitney U test |
| Myostatin (ng/L) | 363 (299–782) | 1108 (388–2655) | 0.001 | Mann–Whitney U test |
| Myonectin (ng/mL) | 2.24 (1.72–5.61) | 5.28 (2.08–13.17) | 0.002 | Mann–Whitney U test |

Data are presented as mean ± standard deviation (SD) for normally distributed variables and as median with interquartile range (IQR; Q1–Q3) for non-normally distributed variables. Between-group comparisons were performed using the independent t test for normally distributed variables and the Mann–Whitney U test for non-normally distributed variables. Categorical variables were compared using the chi-square test. Q1 indicates the 25th percentile and Q3 the 75th percentile. ALP, alkaline phosphatase; 25(OH)D, 25-hydroxyvitamin D; PTH, parathyroid hormone. A p value < 0.05 was considered statistically significant.

**Table 2. Demographic, anthropometric, and biochemical characteristics according to parathyroid hormone (PTH) status.**

| Parameters | Elevated PTH (> 50 pg/mL; n = 37) | Normal PTH (≤ 50 pg/mL; n = 45) | Statistical test |
|---|---|---|---|
| Age (years) | 3.02 (0.95–11.93) | 5.57 (1.63–10.72) | Mann–Whitney U test |
| Sex (n = female/male) | 18/ 19 | 21/ 24 | chi-square test |
| Height SDS | −0.86 ± 1.67 | −0.64 ± 1.43 | independent t test |
| Weight SDS | −0.65 ± 1.61 | −0.64 ± 1.29 | independent t test |
| BMI SDS | 0.19 ± 1.24 | −0.33 ± 0.89 | independent t test |
| Calcium (mg/dL) | 8.1 (6.2–9.1) | 9.9 (9.5–10.3) | Mann–Whitney U test |
| Phosphorus (mg/dL) | 3.7 (2.8–4.8) | 5.3 (4.7–5.6) | Mann–Whitney U test |
| ALP (U/L) | 742 (545–1104) | 229 (184–276) | Mann–Whitney U test |
| Magnesium (mg/dL) | 2.03 ± 0.16 | 2.13 ± 0.17 | independent t test |
| 25(OH)D (ng/mL) | 4.60 (3.19–6.88) | 33.6 (24.05–43.80) | Mann–Whitney U test |
| PTH (pg/mL) | 299 (102–495) | 28.2 (20.15–34.87) | Mann–Whitney U test |
| Irisin (ng/mL) | 14.31 (11.36–45.01) | 24.94 (10.37–69.21) | Mann–Whitney U test |
| Myostatin (ng/L) | 386.4 (318–1029) | 954.3 (306–1995) | Mann–Whitney U test |
| Myonectin (ng/mL) | 2.24 (1.72–5.77) | 4.82 (2.06–11.43) | Mann–Whitney U test |

Data are presented as mean ± standard deviation (SD) for normally distributed variables and as median with interquartile range (IQR; Q1–Q3) for non-normally distributed variables. Comparisons between groups were performed using the independent t test or the Mann–Whitney U test, as appropriate. Categorical variables were compared using the chi-square test. Q1 indicates the 25th percentile and Q3 the 75th percentile. ALP, alkaline phosphatase; 25(OH)D, 25-hydroxyvitamin D. A p value < 0.05 was considered statistically significant.

All statistical tests were two-sided, and a $p < 0.05$ was considered statistically significant. As this was an exploratory clinical study, no a priori power calculation was performed; therefore, the sample size may have limited power to detect moderate effect sizes.

## Results

A total of 43 children with vitamin D deficiency (23 girls and 20 boys; median age 5.46 years [range, 0.17–17.0]) and 39 healthy controls (16 girls and 23 boys; median age 4.03 years [range, 0.31–16.45]) were included in the study. Demographic, anthropometric, and biochemical characteristics of the study groups are summarized in Table 1.

Median serum 25-hydroxyvitamin D [25(OH)D] concentrations were significantly lower in the vitamin D–deficient group than in controls (4.67 ng/mL [IQR 3.39–6.97] vs. 36.1 ng/mL [IQR 27.0–46.7], $p < 0.001$). Compared with controls, the vitamin D–deficient group exhibited significantly higher serum PTH and ALP levels and significantly lower calcium and phosphorus concentrations, with a modest but statistically significant reduction in magnesium levels. Height SDS, weight SDS, and BMI SDS values did not differ significantly between groups (Table 1).

Circulating myokine levels differed significantly between children with vitamin D deficiency and healthy controls. Median serum irisin levels were 13.76 ng/mL (IQR 11.41–34.48) in the vitamin D–deficient group and 29.52 ng/mL (IQR 10.36–70.84) in controls ($p = 0.039$). Similarly, myostatin levels were 363 ng/L (IQR 299–782) versus 1108 ng/L (IQR 388–2655) ($p < 0.001$), and myonectin levels were 2.24 ng/mL (IQR 1.72–5.61) versus 5.28 ng/mL (IQR 2.08–13.17) ($p = 0.002$), respectively (Table 1, Fig 1).

When participants were stratified according to PTH status, 37 children had elevated PTH levels (> 50 pg/mL) and 45 had normal PTH levels (≤ 50 pg/mL). Demographic, anthropometric, and biochemical characteristics of these subgroups are presented in Table 2. In this subgroup analysis, serum myostatin [386.4 ng/L (IQR 318–1029) vs. 954.3 ng/L (IQR 306–1995), $p = 0.033$] and myonectin levels [2.24 ng/mL (IQR 1.72–5.77) vs. 4.82 ng/mL (IQR 2.06–11.43), $p = 0.010$] were significantly lower in children with elevated PTH compared with those with normal PTH levels. In contrast, serum irisin concentrations did not differ significantly between PTH groups [14.31 ng/mL (IQR 11.36–45.01) vs. 24.94 ng/mL (IQR 10.37–69.21), $p = 0.164$] (Table 2, Fig 2). In the elevated PTH group, serum calcium, phosphorus, and magnesium levels were lower, and ALP levels were higher.

Correlation analyses were performed separately for the vitamin D–deficient and control groups, and $p$ values were adjusted for multiple comparisons using the Benjamini–Hochberg false discovery rate (FDR) method (Table 3).

In the vitamin D–deficient group, no significant correlations were observed between circulating myokine levels and serum 25(OH)D, PTH, anthropometric indices, or most biochemical parameters after FDR adjustment. However, strong positive correlations were identified among the three circulating myokines, including associations between log-transformed

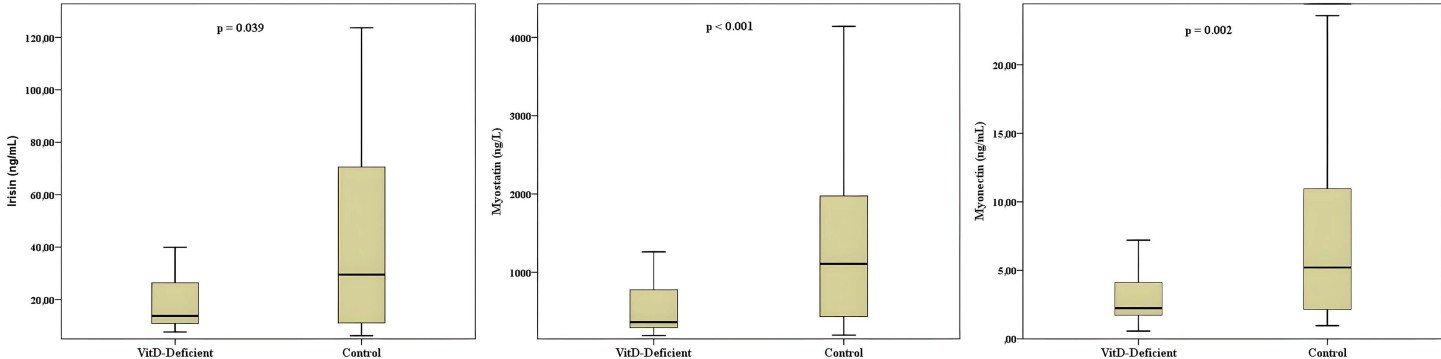

**Fig 1. Comparison of serum irisin, myostatin, and myonectin levels between vitamin D-deficient and control groups.** Boxplots display median and interquartile range (Q1–Q3), and whiskers represent minimum–maximum values. Group differences were observed for all three myokines (irisin: $p = 0.039$; myostatin: $p < 0.001$; myonectin: $p = 0.002$). Group differences were evaluated using the Mann–Whitney U test, as all variables showed non-normal distribution.

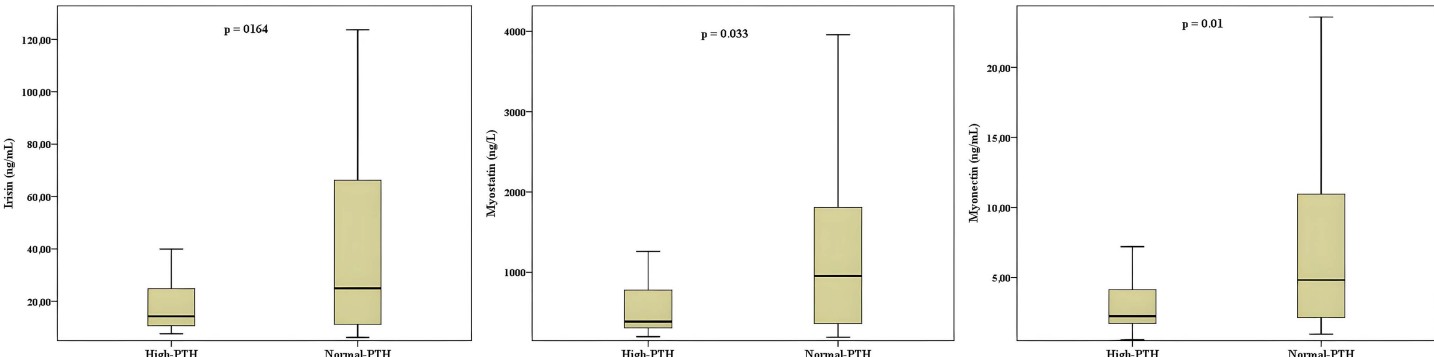

**Fig 2. Comparison of serum irisin, myostatin, and myonectin levels between children with high PTH (>50 pg/mL) and normal PTH (≤50 pg/mL).** Boxplots display median and interquartile range (Q1–Q3), and whiskers represent minimum–maximum values. Group differences were observed for myostatin ($p = 0.033$) and myonectin ($p = 0.010$), whereas no significant difference was observed for irisin ($p = 0.164$). Group differences were evaluated using the Mann–Whitney U test, as all variables showed non-normal distribution.

**Table 3. Correlation between quantitative variables and circulating myokines in vitamin D–deficient and control groups after adjustment for multiple comparisons using the Benjamini–Hochberg false discovery rate method.**

| Variables | Vitamin D-deficient group | | | Control group | | |
|---|---|---|---|---|---|---|
| | Ln(Irisin) | Ln(Myostatin) | Ln(Myonectin) | Ln(Irisin) | Ln(Myostatin) | Ln(Myonectin) |
| Weight SDS | −0.250[x] (0.242) | −0.144[x] (0.520) | −0.094[y] (0.747) | 0.092[y] (0.733) | 0.134[y] (0.548) | 0.143[y] (0.533) |
| Height SDS | −0.300[x] (0.239) | −0.156[x] (0.520) | −0.080[y] (0.747) | 0.109[y] (0.733) | 0.150[y] (0.548) | 0.147[y] (0.533) |
| BMI SDS | −0.091[x] (0.620) | −0.098[x] (0.662) | −0.103[y] (0.747) | −0.002[y] (0.989) | 0.053[y] (0.770) | 0.047[y] (0.793) |
| Calcium | −0.255[x] (0.239) | −0.355[x] (0.086) | −0.077[x] (0.747) | 0.418[x] (0.016) | 0.348[x] (0.060) | 0.376[x] (0.041) |
| Phosphorus | −0.101[x] (0.588) | 0.002[x] (0.988) | −0.101[y] (0.747) | 0.553[y] (<0.001) | 0.593[y] (<0.001) | 0.553[y] (0.002) |
| ALP | 0.248[x] (0.239) | 0.314[x] (0.086) | 0.161[x] (0.747) | 0.141[x] (0.615) | 0.171[x] (0.495) | 0.150[x] (0.533) |
| Magnesium | −0.159[x] (0.472) | −0.191[x] (0.493) | −0.246[y] (0.747) | 0.578[y] (0.002) | 0.581[y] (0.002) | 0.533[y] (0.009) |
| PTH | 0.237[x] (0.239) | 0.316[x] (0.086) | 0.107[x] (0.747) | −0.061[y] (0.795) | −0.118[y] (0.548) | −0.079[y] (0.729) |
| 25(OH)D | −0.209[x] (0.245) | −0.329[x] (0.086) | −0.051[x] (0.747) | 0.415[x] (0.016) | 0.366[x] (0.055) | 0.409[x] (0.029) |
| Ln(Myostatin) | 0.847[x] (<0.001) | – | – | 0.961[y] (<0.001) | – | – |
| Ln(Myonectin) | 0.814[x] (<0.001) | 0.729[x] (<0.001) | – | 0.960[y] (<0.001) | 0.972[y] (<0.001) | – |

Data are presented as correlation coefficients ($r$) and FDR-adjusted $p$ values in the format of $r$ (adjusted $p$). Circulating myokine concentrations were log-transformed prior to correlation analyses. [x] Spearman's rank correlation coefficient; [y] Pearson correlation coefficient. Ln, natural logarithm; SDS, standard deviation score; BMI, body mass index; PTH, parathyroid hormone; ALP, alkaline phosphatase; 25OHD, 25-hydroxyvitamin D. $p$ values were adjusted for multiple comparisons using the Benjamini–Hochberg false discovery rate (FDR) method.

irisin and log-transformed myostatin ($r = 0.847$, adjusted $p < 0.001$), irisin and myonectin ($r = 0.814$, adjusted $p < 0.001$), and myostatin and myonectin ($r = 0.729$, adjusted $p < 0.001$).

In the control group, serum 25(OH)D concentrations showed positive correlations with circulating irisin ($r = 0.415$, adjusted $p = 0.016$) and myonectin ($r = 0.409$, adjusted $p = 0.029$), whereas the correlation with myostatin did not remain statistically significant after adjustment. In addition, phosphorus and magnesium levels were positively correlated with all three circulating myokines. Similar to the vitamin D–deficient group, strong positive correlations were observed among the circulating myokines themselves, including associations between irisin and myostatin ($r = 0.961$, adjusted $p < 0.001$), irisin and myonectin ($r = 0.960$, adjusted $p < 0.001$), and myostatin and myonectin ($r = 0.972$, adjusted $p < 0.001$) (Table 3).

### Primary multivariable models

To evaluate the independent association between vitamin D status and circulating myokines, multivariable linear regression analyses were performed adjusting for age, sex, and BMI SDS. Serum 25(OH)D remained independently associated with log-transformed irisin (β = 0.014, p = 0.023), log-transformed myostatin (β = 0.016, p = 0.012), and log-transformed myonectin (β = 0.021, p = 0.006). Variance inflation factor (VIF) values ranged between 1.0 and 1.1, indicating no evidence of significant multicollinearity among predictors (S1 File).

### Multiple-comparison sensitivity

Given the number of biochemical correlations tested, sensitivity analyses were performed using the false discovery rate (FDR) correction. After adjustment, the associations between 25(OH)D and myostatin and myonectin remained significant, whereas the association with irisin was attenuated after correction (S2 Table in S1 File).

### Secondary PTH sensitivity analysis

In secondary sensitivity models including log-transformed PTH in addition to age, sex, BMI SDS, and 25(OH)D, the association between 25(OH)D and circulating myokines remained significant. Log(PTH) was independently associated with irisin but not with myostatin or myonectin. VIF values remained well below the predefined threshold (all < 2.5), suggesting acceptable collinearity.

## Discussion

Given the cross-sectional and exploratory design of this study, the findings should be interpreted as associative rather than causal. Within this framework, the present study shows that circulating myokine levels are reduced in children with vitamin D deficiency and that myostatin and myonectin levels are also lower in those with elevated PTH concentrations. In addition, the correlations observed between serum 25(OH)D levels and selected circulating myokines—particularly irisin and myonectin in the control group, whereas no significant correlations were observed in the vitamin D–deficient group— suggest that vitamin D status may be associated with alterations in skeletal muscle–derived endocrine markers.

Importantly, the observed associations persisted after adjustment for age, sex, and BMI SDS in multivariable regression analyses and were largely robust to sensitivity analyses with multiple-comparison correction, reinforcing the independent association between vitamin D status and circulating myokines in this pediatric cohort.

Skeletal muscle expresses the vitamin D receptor (VDR) and possesses local 1α-hydroxylase activity, enabling responsiveness to vitamin D–related signaling pathways [9]. Experimental studies have suggested that vitamin D–related pathways may be involved in muscle metabolic processes [12]. However, VDR expression or activity was not assessed in the present study, and therefore no conclusions regarding receptor-mediated mechanisms can be drawn. In adults, low circulating levels of active vitamin D [1,25(OH)$_2$D] have been associated with reduced muscle strength, increased inflammatory markers, and impaired fatty acid oxidation, with improvements observed following exercise interventions [23]. Although these data originate from adult populations and heterogeneous clinical contexts and may not be directly comparable to children, they provide biological plausibility for the associations observed in the present pediatric cohort. In this context, the observed reductions in circulating myokine levels may reflect altered muscle-related biochemical profiles associated with vitamin D deficiency during childhood, a period characterized by rapid growth and dynamic musculoskeletal development. Within this broader musculoskeletal framework, bone and skeletal muscle are increasingly recognized as interconnected tissues that communicate through mechanical, biochemical, and endocrine pathways, forming a functional bone–muscle unit that is particularly relevant during growth and development [11].

Serum irisin concentrations were lower in children with vitamin D deficiency and showed a positive correlation with 25(OH)D levels. Previous studies have reported increases in circulating irisin following vitamin D supplementation, accompanied by improvements in metabolic parameters [15,24,25]. Similar findings have been described in children with Prader–Willi

syndrome receiving vitamin D replacement [25]. In the present study, lower irisin levels may reflect alterations in muscle-derived metabolic signaling, processes that are relevant to growth, body composition, and metabolic regulation during childhood.

Myostatin, a negative regulator of myogenesis, is typically elevated under catabolic conditions. In contrast, circulating myostatin levels were lower in children with vitamin D deficiency in the present study. This observation may reflect developmental or metabolic characteristics of pediatric muscle regulation, a period characterized by a predominance of anabolic pathways. Previous studies in adults have reported inconsistent associations between vitamin D status and myostatin levels, with both positive and negative correlations described [26–29]. Such discrepancies may be related to differences in age, physical activity, nutritional status, and metabolic context. Pediatric-specific regulatory mechanisms may therefore modulate myostatin expression differently from those observed in adult populations. It is also important to recognize that circulating myostatin concentrations may not directly reflect intramuscular myostatin signaling or functional muscle mass. Myostatin is part of a broader regulatory network involving bone-derived and endocrine mediators within the bone–muscle unit [10,11]. Experimental evidence suggests that vitamin D–related pathways can influence muscle anabolic and catabolic balance through modulation of oxidative stress, mitochondrial function, and metabolic signaling cascades [12]. Within this integrated framework, alterations in circulating myostatin in vitamin D–deficient children may represent systemic adaptive responses rather than isolated changes in muscle catabolism. Therefore, the lower myostatin levels observed in our cohort should be interpreted cautiously and not as direct evidence of enhanced muscle growth.

Myonectin, a myokine involved in glucose utilization and lipid oxidation within the muscle–liver axis, was also reduced in children with vitamin D deficiency, particularly among those with concomitant secondary hyperparathyroidism. This finding is consistent with an association between vitamin D status, PTH levels, and myonectin concentrations. To our knowledge, this study is among the first to report such an association in a pediatric population, highlighting potential metabolic correlates during growth.

In pediatric endocrine practice, vitamin D deficiency is frequently accompanied by high bone turnover and accelerated skeletal remodeling. These growth-related skeletal dynamics may further influence muscle–bone endocrine cross-talk during development. Vitamin D deficiency is commonly accompanied by secondary hyperparathyroidism, and increased PTH secretion may further influence muscle endocrine activity. In the present cohort, lower 25(OH)D concentrations in the high-PTH group indicate that elevated PTH largely reflects more severe vitamin D deficiency. Although PTH is classically considered to act on bone and kidney, experimental data indicate that PTH receptors are also expressed in skeletal muscle [6,30]. Previous studies suggest that the effects of PTH on muscle may vary according to exposure patterns and metabolic context [6,30,31]. In this study, the observation of lower myostatin and myonectin levels in children with elevated PTH concentrations is consistent with differences in myokine profiles according to PTH status, although a direct suppressive effect cannot be inferred due to the cross-sectional design.

Indirect interactions between bone and muscle may also contribute to the observed findings. Increased bone resorption associated with vitamin D deficiency and elevated PTH levels may lead to the release of bone-derived factors, such as transforming growth factor-β (TGF-β), which has been shown to exert paracrine effects on skeletal muscle [32,33]. Although bone turnover markers and TGF-β levels were not assessed in the present study, these pathways provide a plausible biological context in which alterations in circulating myokine levels may, at least in part, reflect indirect effects mediated by bone metabolism rather than solely direct actions on muscle tissue.

The relationship between PTH and irisin remains inconsistent in the literature. Reduced irisin levels have been reported in adults with primary hyperparathyroidism [34], whereas positive associations have been described in pediatric cancer survivors [35]. In the present study, no significant correlation between PTH and irisin was observed. This finding may reflect differences in study populations and analytical power. Given the exploratory nature of the analyses, these results should be interpreted cautiously and considered hypothesis-generating. At present, no established reference ranges or validated threshold concentrations defining normal skeletal muscle function exist for circulating irisin, myostatin, or myonectin, particularly in pediatric populations. Therefore, the findings of this study should be interpreted as relative differences between groups rather than absolute indicators of muscle dysfunction.

Several biological and contextual factors may contribute to variability in circulating myokine levels in pediatric populations. Physical activity is known to influence myokine secretion, particularly irisin and myonectin, through exercise-related muscle activity [14]. Nutritional status, including protein intake and overall energy balance, may also affect muscle metabolism during growth, as dietary patterns and protein quality have been shown to influence circulating myokine concentrations and muscle metabolic processes in children [16]. In addition, pubertal maturation is characterized by hormonal changes that could interact with muscle-derived endocrine signaling [17]. Although these variables were not systematically assessed, their potential influence should be considered when interpreting the observed associations.

Circulating myokines were measured using ELISA-based assays. Despite reporting assay performance characteristics, variability related to assay sensitivity and the lack of standardized pediatric reference ranges may limit direct clinical interpretation [36]. Therefore, the findings should be interpreted as relative differences within this cohort rather than definitive indicators of muscle dysfunction.

## Strengths and limitations

The main strength of this study is the simultaneous evaluation of three key myokines—irisin, myostatin, and myonectin—in relation to both vitamin D and PTH status in a pediatric population. Several limitations should be acknowledged. First, the cross-sectional design precludes causal inference, and bidirectional associations cannot be excluded. Second, no a priori power calculation was performed, and the study may have been underpowered for certain subgroup analyses; moreover, given the number of biochemical variables examined, the risk of type I error cannot be entirely excluded, although sensitivity analyses with multiple-comparison correction supported the robustness of the primary findings. Regression analyses were conducted using complete-case data, which may introduce some degree of selection bias; however, no substantial differences in baseline characteristics were observed between included and excluded participants. Third, relevant determinants of muscle endocrine function—including physical activity, pubertal stage, body composition, and seasonal variation—were not systematically assessed. Fourth, circulating myokines were measured using ELISA-based assays, and the lack of standardized pediatric reference ranges limits clinical interpretability. Finally, functional muscle assessments and tissue-level analyses were not performed, restricting mechanistic conclusions, and the findings may not be fully generalizable to milder or asymptomatic forms of vitamin D deficiency.

## Conclusions

This study shows that vitamin D deficiency in childhood is associated with reduced circulating levels of irisin, myostatin, and myonectin, indicating alterations in skeletal muscle endocrine activity. The observed associations between vitamin D status, parathyroid hormone levels, and myokine profiles are consistent with a possible involvement of the vitamin D–PTH axis in muscle–bone and metabolic interactions during growth. Although the cross-sectional design precludes causal inference, these findings underscore the potential relevance of maintaining adequate vitamin D status in children in relation to musculoskeletal development and metabolic health, and support the need for further longitudinal and interventional studies.

## Supporting information

**S1 File. Supplementary tables including robust and multivariable regression analyses (S1 and S2 Tables).**
(DOCX)

## Author contributions

**Conceptualization:** Muammer Buyukinan.

**Data curation:** Huseyin Kurku, Yavuz Turgut Gederet, Ahmet Fatih Yilmaz.

**Formal analysis:** Muammer Buyukinan, Zafer Bagci, Ahmet Fatih Yilmaz.

**Investigation:** Huseyin Kurku, Yavuz Turgut Gederet.

**Methodology:** Muammer Buyukinan.

**Software:** Ahmet Fatih Yilmaz.

**Supervision:** Muammer Buyukinan.

**Writing – original draft:** Muammer Buyukinan.

**Writing – review & editing:** Muammer Buyukinan, Zafer Bagci.

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
