## [Decision Letter · Decision Letter 0]

18 Feb 2026

PONE-D-26-03690

Association between vitamin D status and circulating myokines (irisin, myostatin, and myonectin) in children: a cross-sectional study

PLOS One

Dear Dr. Buyukinan,

Thank you for submitting your manuscript to PLOS ONE. After careful consideration, we feel that it has merit but does not fully meet PLOS ONE’s publication criteria as it currently stands. Therefore, we invite you to submit a revised version of the manuscript that addresses the points raised during the review process.

We look forward to receiving your revised manuscript.

Kind regards,

Esedullah Akaras

Academic Editor

PLOS One

**Journal Requirements:**

Reviewers' comments:

Reviewer's Responses to Questions

**Comments to the Author**

1. Is the manuscript technically sound, and do the data support the conclusions?

Reviewer #1: Yes

Reviewer #2: Partly

2. Has the statistical analysis been performed appropriately and rigorously?

Reviewer #1: Yes

Reviewer #2: Yes

3. Have the authors made all data underlying the findings in their manuscript fully available?

Reviewer #1: Yes

Reviewer #2: Yes

4. Is the manuscript presented in an intelligible fashion and written in standard English?

Reviewer #1: Yes

Reviewer #2: Yes

5. Review Comments to the Author

Reviewer #1: Rewrite / polish the Discussion (especially myostatin & limitations)

Add a strong Limitations paragraph (PLOS One–style)

Prepare a formal “Response to Reviewers” letter

Language editing + scientific tone polishing for the whole manuscript

Improve the Abstract (results & conclusions)

Reviewer #2: Overview comments

I commend the authors of this Manuscript titled “Association between vitamin D status and circulating myokines (irisin, myostatin, and myonectin) in children: a cross-sectional study “. This study explores a timely question on how vitamin D status relates to muscle-derived hormones (myokines) in children. However, I would encourage the authors to address a few gaps pointed in various sections to strengthen their work and contribute more robustly to the field.

Introduction

• The manuscript addresses a gap, but there is a need to expand the literature review by citing more recent systematic reviews and clarify why irisin, myostatin, and myonectin were chosen. Could you link the biological functions of irisin, myostatin, and myonectin to existing vitamin D pathways to provide a stronger theoretical framework.

• As much as the confounders like physical activity, nutrition and pubertal status are mentioned as limitations, there is a need for their deeper discussion in the introduction to strengthen the rationale and their potential impact in pediatric populations.

Methods

• Owing to the numerous biochemical variables in your study, it is important to assess multicollinearity among these variables to cater for correlation bias. I would advise you report correlation matrices or Variance Inflation Factor (VIF) if possible, in the supplementary material.

Results

• The main finding that children with vitamin D deficiency have lower levels of irisin, myostatin and myonectin is novel.

• The presentation of correlation analyses is well done, but without adjustment for multiple comparisons or confounders, the risk of false positives could arise. I suggest you apply a correction like Bonferroni for significant p-values to increase the robustness of the findings. If not possible to report correction for multiple comparisons, acknowledge the risk of type I error due to the high number of biochemical variables.

Discussion

• The discussion appropriately acknowledges the cross-sectional design’s limitations and the need for longitudinal studies.

• Some limitations are acknowledged, but assay validation and confounder control need more emphasis. I propose that the authors consider expanding the discussion on how unmeasured confounders like physical activity and nutritional status might have influenced the observed myokine levels, even though they have been noted as limitations.

Conclusions

• They tie to the results but causal language could be avoided

Readability

• Line numbering is key in enhancing readability and ease of reference. This is missing

6. PLOS authors have the option to publish the peer review history of their article (what does this mean?). If published, this will include your full peer review and any attached files.

**Do you want your identity to be public for this peer review?** For information about this choice, including consent withdrawal, please see our Privacy Policy.

Reviewer #1: **Yes:**dr firouzeh dehghan

Reviewer #2: No

---

## [Author Response · Author response to Decision Letter 1]

16 Mar 2026

We thank the editor and the reviewers for their constructive and insightful comments.

All reviewer comments have been carefully addressed and the manuscript has been revised accordingly.

Detailed point-by-point responses are provided in the "Response to Reviewers" document, and all corresponding changes have been incorporated into the revised manuscript.

We hope that the revised version will now be suitable for publication in PLOS ONE.

---

## [Decision Letter · Decision Letter 1]

23 Apr 2026

Association between vitamin D status and circulating myokines (irisin, myostatin, and myonectin) in children: a cross-sectional study

PONE-D-26-03690R1

Dear Dr. Buyukinan,

We’re pleased to inform you that your manuscript has been judged scientifically suitable for publication and will be formally accepted for publication once it meets all outstanding technical requirements.

Kind regards,

Esedullah Akaras

Academic Editor

PLOS One

Additional Editor Comments (optional):

Reviewers' comments:

Reviewer's Responses to Questions

**Comments to the Author**

1. If the authors have adequately addressed your comments raised in a previous round of review and you feel that this manuscript is now acceptable for publication, you may indicate that here to bypass the “Comments to the Author” section, enter your conflict of interest statement in the “Confidential to Editor” section, and submit your "Accept" recommendation.

Reviewer #1: All comments have been addressed

Reviewer #2: (No Response)

2. Is the manuscript technically sound, and do the data support the conclusions?

Reviewer #1: Yes

Reviewer #2: Yes

3. Has the statistical analysis been performed appropriately and rigorously? 

Reviewer #1: Yes

Reviewer #2: Yes

4. Have the authors made all data underlying the findings in their manuscript fully available?

Reviewer #1: Yes

Reviewer #2: Yes

5. Is the manuscript presented in an intelligible fashion and written in standard English?

Reviewer #1: Yes

Reviewer #2: Yes

6. Review Comments to the Author

Reviewer #1: accept with this format

Please use the space provided to explain your answers to the questions above. You may also include additional comments for the author, including concerns about dual publication, research ethics, or publication ethics. (Please upload your review as an attachment if it exceeds 20,000 characters) (Limit 100 to 20000 Characters)

Minimum Character Count Not Met

Reviewer #2: I have reviewed the revised manuscript (PONE-D-26-03690_R1) and the authors' detailed responses to my previous comments. The manuscript has been significantly improved with a clearer theoretical framework for selecting specific myokines and the application of more robust statistical methods, including the Benjamini–Hochberg False Discovery Rate correction for multiple comparisons.

The addition of a discussion on unmeasured confounders, such as physical activity and pubertal status, provides essential context for interpreting the pediatric findings. The authors have also appropriately adjusted the tone of the manuscript to reflect its cross-sectional nature, avoiding unwarranted causal inferences.

The inclusion of supplementary materials (S1 and S2 Tables, covering VIF and sensitivity analyses) is a valuable enhancement. However, the term "dependent" in S2 Table needs to be revised for accuracy.

7. PLOS authors have the option to publish the peer review history of their article (what does this mean?). If published, this will include your full peer review and any attached files.

Reviewer #1: No

Reviewer #2: **Yes:**Kinuthia Stanley

---

## [Editor Report · Acceptance letter]

PONE-D-26-03690R1

PLOS One

Dear Dr. Buyukinan,

I'm pleased to inform you that your manuscript has been deemed suitable for publication in PLOS One. Congratulations! Your manuscript is now being handed over to our production team.

Kind regards,

on behalf of

Dr. Esedullah Akaras

Academic Editor

PLOS One